# Genomic Diversity of NDM-Producing *Klebsiella* Species from Brazil, 2013–2022

**DOI:** 10.3390/antibiotics11101395

**Published:** 2022-10-12

**Authors:** Carlos Henrique Camargo, Amanda Yaeko Yamada, Andreia Rodrigues de Souza, Alex Domingos Reis, Marlon Benedito Nascimento Santos, Denise Brandão de Assis, Eneas de Carvalho, Elizabeth Harummyy Takagi, Marcos Paulo Vieira Cunha, Monique Ribeiro Tiba-Casas

**Affiliations:** 1Centro de Bacteriologia, Instituto Adolfo Lutz, São Paulo 01246-902, Brazil; 2Faculdade de Medicina, Universidade de São Paulo, São Paulo 01246-903, Brazil; 3Laboratório Estratégico, Instituto Adolfo Lutz, São Paulo 01246-000, Brazil; 4Divisão de Infecção Hospitalar, Centro de Vigilância Epidemiológica, São Paulo 01246-902, Brazil; 5Department of Biotechnology (NuCEL), Instituto Butantan, São Paulo 05503-900, Brazil; 6Núcleo de Coleção de Micro-Organismos, Instituto Adolfo Lutz, São Paulo 01246-000, Brazil; 7School of Veterinary Medicine and Animal Science, Universidade de São Paulo, São Paulo 05508-270, Brazil

**Keywords:** *Klebsiella*, whole genome sequencing, MLST, CG258, NDM-1, NDM-7, Illumina, MinIon, ONT, polymyxin B

## Abstract

**Background:** Since its first report in the country in 2013, NDM-producing Enterobacterales have been identified in all the Brazilian administrative regions. In this study, we characterized by antimicrobial susceptibility testing and by molecular typing a large collection of NDM-producing *Klebsiella* isolates from different hospitals in Brazil, mainly from the state of Sao Paulo, over the last decade. **Methods**: Bacterial isolates positive for *bla*_NDM_-genes were identified by MALDI-TOF MS and submitted to antimicrobial susceptibility testing by disk diffusion or broth microdilution (for polymyxin B). All isolates were submitted to pulsed-field gel electrophoresis, and isolates belonging to different clusters were submitted to whole genome sequencing by Illumina technology and downstream analysis. Mating out assays were performed by conjugation, plasmid sizes were determined by S1-PFGE, and plasmid content was investigated by hybrid assembly after MinIon long reads sequencing. **Results**: A total of 135 NDM-producing *Klebsiella* were identified, distributed into 107 different pulsotypes; polymyxin B was the only antimicrobial with high activity against 88.9% of the isolates. Fifty-four isolates presenting diversified pulsotypes were distributed in the species *K. pneumoniae* (70%), *K. quasipneumoniae* (20%), *K. variicola* (6%), *K. michiganensis* (a *K. oxytoca* Complex species, 2%), and *K. aerogenes* (2%); *bla*_NDM-1_ was the most frequent allele (43/54, 80%). There was a predominance of Clonal Group 258 (ST11 and ST340) encompassing 35% of *K. pneumoniae* isolates, but another thirty-one different sequence types (ST) were identified, including three described in this study (ST6244 and ST6245 for *K. pneumoniae*, and ST418 for *K. michiganensis*). The *bla*_NDM-1_ and *bla*_NDM-7_ were found to be located into IncF and IncX3 type transferable plasmids, respectively. **Conclusions**: Both clonal (mainly driven by CG258) and non-clonal expansion of NDM-producing *Klebsiella* have been occurring in Brazil in different species and clones, associated with different plasmids, since 2013.

## 1. Introduction

Antimicrobial resistance remains a challenge for public health, with direct impacts on increased mortality, length of hospitalization stay, and economic losses, especially in low- and middle-income countries [1]. In Brazil, high and endemic rates of bacterial resistance are reported across different health care settings [2,3], remarkably among Gram-negative pathogens. In *Klebsiella*, carbapenem resistance is mainly mediated by the production of the *Klebsiella pneumoniae* Carbapenemase (KPC) enzyme, but emerging and infrequent carbapenemases have also been reported [4]. The New Delhi Metallo-β-lactamase (NDM) is a potent carbapenemase, initially identified in the Indian subcontinent [5] and now widespread in many countries [6]. In Brazil, the first NDM-producing bacteria were identified in the southern region in 2013 [7], and since then, they have been reported in all five Brazilian administrative regions from both clinical and surveillance specimens [8]. In addition, this increased number of NDM-producing isolates reported in Brazil, limited data on their genetic diversity is available, especially regarding their clonal structure and the diversity of *bla*_NDM_ alleles [8,9]. To elucidate these unknown issues of NDM-producing *Klebsiella* spp. from Brazil, we evaluated the clonal relatedness and the antimicrobial susceptibility of a collection of isolates recovered from different hospitals between 2013 and 2022 and received by our reference laboratory.

## 2. Results

During 2013–2022, we identified 135 cases of *Klebsiella* isolates with positive PCR results for the *bla*_NDM_ gene. Those isolates were recovered from surveillance (rectal or perianal) swab (n = 67), urine (n = 25), blood or catheter tip (n = 23), upper respiratory tract secretion (n = 12), cerebrospinal fluid (CSF) (n = 2), and other clinical samples (n = 6) from non-repetitive patients. These isolates were mainly recovered from hospitals in the states of the Southeast Region (n = 109; 80.7%) but were also representative of the North (n = 21; 15.5%), Northeast (n = 4; 3.7%), and Midwest (n = 1; 0.7%) Regions (Appendix A).

All the 135 isolates were submitted to antimicrobial susceptibility testing against 20 antimicrobial agents, and only amikacin and polymyxin B presented activity against more than 50% of the isolates (56.3% and 88.9% of susceptibility, respectively) (Table 1). Most of the isolates (n = 106; 78.5%) presented multiple antimicrobial resistance (AMR) index ≥0.8 (Figure 1). Since polymyxin B susceptibility was determined by broth microdilution, the MIC50 and MIC90 values could be determined as 0.5 μg/mL and 4.0 μg/mL, respectively. These 135 isolates were also typed by pulsed-field gel electrophoresis (PFGE), and they clustered into 107 different pulsotypes (Simpson Index = 0.995) (Appendix A).

Based on the main clusters identified by PFGE, 54 isolates were selected for whole genome sequencing (WGS) (as indicated in Appendix A). They were representatives of seven different states from four Brazilian administrative regions (Appendix A). These isolates were identified as *K. pneumoniae* (n = 38; 70%), *K. quasipneumoniae* (n = 11; 20%), *K. variicola* (n = 3; 6%), *K. michiganensis* (a *K. oxytoca* Complex species, n = 1; 2%), and *K. aerogenes* (n = 1; 2%). In the whole genome sequencing analysis, the *bla*_NDM_ allele could be determined as *bla*_NDM-1_ (n = 46; 85%) or *bla*_NDM-7_ (n = 5; 9%); intriguingly, in three isolates, the *bla*_NDM_ gene could not be identified in WGS even with the raw data (FASTQ files), but they persisted positive in conventional PCR carried out with the same DNA sample extracted for WGS.

A total of 33 different STs were identified among the 54 *Klebsiella* isolates. *K. aerogenes* was identified as ST128 and *K. michiganensis* as a new sequence type, ST418. Three different STs were identified among the *K. variicola* isolates, each one represented by one isolate: ST1456, ST2586, and ST4609. For *K. quasipneumoniae*, the 11 isolates were identified as seven different STs: ST196 (three isolates); ST283 and ST1040 (two isolates, each); and ST367, ST477, ST526, and ST1822 (one isolate, each). For the 38 *K. pneumoniae* isolates, 21 different STs were identified, with ST11 and ST340 (belonging to Clonal Group 258) accounting for 13 isolates; ST15 and ST392 for three isolates each; and ST147 and ST6245 (new) for two isolates each; the remaining 15 isolates were classified into 15 different STs, including another one described in this study: ST16, ST17, ST37, ST43, ST281, ST307, ST336, ST395, ST460, ST464, ST485, ST534, ST874, ST3128, and ST6244 (new) (Appendix A).

The phylogenetic tree based on single nucleotide polymorphisms (SNP) (Figure 1) showed different branches correlating well with species identification but contrasting with the inability of PFGE to cluster the different *Klebsiella* species apart from each other (Appendix A). Pairwise comparison of each genome identified that the SNPs variation ranged from 1 to 50,489. We observed a main cluster comprising the *K. pneumoniae* isolates with SNPs ranging from 1 to 9639. The *K. quasipneumoniae* cluster was subdivided in two subclusters with SNPs ranging from 11 to 9186 and from 78 to 9070, respectively; the overall distance among *K. quasipneumoniae* was 29,757 SNPs. The highest diversity was observed for *K. variicola*, with SNP numbers ranging from 9185 to 9899. In this tree (Figure 1), we observe the distribution of the main β-lactam-encoding genes (extended spectrum β-lactamase – ESBL, and carbapenemases) as well as some virulence genes associated with siderophores production (*ent*, *ybt*, *iuc*, *iroB*, *iroN*), colibactin (*clb*), and hypermucoviscosity phenotype (*rmpA*/*A2*). CTX-M-15 was the main ESBL detected (in 76% of the isolates) and six isolates (11%) were also KPC-2 co-producers. The *ybt* operon, which codifies for yersiniabactin siderophores, was found in 17 *K. pneumoniae* isolates (31%), most of them of the ST11 (12/17). On the other hand, *clb* and *rmpA*/*A2* genes were not detected, while *iroB* and *iroN* were detected only in the *K. aerogenes* isolate.

After the ST and *bla*_NDM_-alleles determination, two isolates with *bla*_NDM-1_ (*K. pneumoniae* ST15 and ST460) and three with *bla*_NDM-7_ (*K. pneumoniae* ST11, ST147; *K. quasipneumoniae* ST1822) were submitted to mating-out assays, S1-PFGE, and long read genome sequencing for further plasmid characterization. Hybrid assembly allowed the identification of the *bla*_NDM-1_ gene in IncF-type plasmids, while *bla*_NDM-7_ was identified in IncX3 plasmids. Pairwise alignment of the IncX3-*bla*_NDM-7_ plasmids is presented in Figure 2. NDM activity was confirmed in the transconjugant strains, both in NDM-1 and NDM-7, including a >32-fold increase in carbapenems’ minimal inhibitory concentrations (Table 2).

## 3. Discussion

We conducted the largest Brazilian laboratory-based surveillance study on NDM-producing *Klebsiella* species with isolates collected from different hospitals over the last 10 years. Besides the high prevalence of *K. pneumoniae* isolates from the Clonal Group CG258 (represented by ST11 and ST340), a pronounced genetic diversity was observed, indicating that dissemination of this genetic determinant has also been driven by non-clonal expansion.

*K. pneumoniae* is recognized as the main pathogen associated with NDM production in Brazil [8], and 20 different STs were identified among the 38 isolates presenting diversified PFGE pulsotypes, including two new STs resulting from new allele combinations (ST6244 and ST6245). This study identified other 14 STs that have never been associated with NDM carbapenemase in our country, providing an update to the molecular epidemiological scenario of NDM-producing *K. pneumoniae* in Brazil, a continental country that suffers from high and endemic levels of antimicrobial resistance among pathogens causing hospital acquired infections [2,3]. In a recent narrative review of NDM-producing Enterobacterales in Brazil, we identified that *bla*_NDM-1_ is mainly identified in *Klebsiella* species, belonging to ST350 and ST15 [8].

Globally, NDM is found in a diversified bacterial genetic background around the world, including in the epidemic clones *K. pneumoniae* ST11 and ST147 [6,10,11]. In this study, we identified globally recurrent clones (ST11, ST147, ST340, and ST15) in our isolates, reflecting the dynamics of antimicrobial resistance observed abroad [10]. CG258 (which comprises ST11 and ST340) is one of the most disseminated clones worldwide, usually associated with multidrug antimicrobial resistance [12]. We found that NDM is also becoming frequent in this high-risk clone. Besides *K. pneumoniae*, we also detected NDM-producing clones of *K. variicola* and *K. quasipneumoniae* belonging to STs never identified before in the Brazilian territory, indicating that less frequent pathogens can also carry *bla*_NDM_ genes. Among the 21 STs identified in our study, at least seven of them have already been associated with outbreaks worldwide: ST11 (Bulgaria, Greece, and Turkey), ST15 (Nepal), ST17 and ST37 (China), ST147 (China and Tunisia), ST392 (Mexico), and ST340, which caused a large outbreak in two Brazilian hospitals [11,13]. These findings indicate that occasional clones have the potential to cause outbreaks under favorable conditions.

For the first time in America, we identified the occurrence of the *bla*_NDM-7_ gene located in a transferable IncX3 plasmid occurring in *K. pneumoniae* and *K. quasipneumoniae* isolates. The high similarity observed in the structures of these plasmids suggests that plasmid spread can have occurred, since it was identified in different lineages of *K. pneumoniae* (ST11 and ST147) and even in another species. In contrast, *bla*_NDM-1_ was found to be located in IncF plasmids in our isolates. A recent study found the occurrence of *bla*_NDM-1_ in a megaplasmid carried by a *K. pneumoniae* ST1588 recovered from rectal swab in a patient from Chile [14]. Since 2011, a variety of plasmids have been found to be associated with the NDM gene, such as the incompatibility (Inc) groups IncF, IncA/C, IncL/M, or even on the bacterial chromosome [15]. In Brazil, the *bla*_NDM_ gene was also observed in plasmids with different sizes [16], indicating that this resistance determinant occurs in different mobile genetic element backbones [11].

Both NDM-1- and NDM-7-producing *Klebsiella* species presented high co-resistances to several antimicrobials, while polymyxin activity remained preserved against almost 90% of the evaluated isolates. Although polymyxin resistance in carbapenemase producing bacteria has been increasing in some hospitals with high endemic KPC rates [17], our findings indicate that this is not a reality for NDM-producing bacteria, at least until now. Antimicrobial therapy of multidrug-resistant pathogens is a challenge for medical practice, particularly in institutions with limited access to new antimicrobial options, and, unfortunately, even the new drug combinations (ceftazidime-avibactam, imipenem-relebactam, and meropenem-vaborbactam) are ineffective against MBL-producing bacteria [18]. In fact, a study showed that in a Greek hospital in which ceftazidime-avibactam was largely employed, a reversal in the carbapenemase epidemiology was observed towards MBL enzymes [19].

Since the development, approval, and release of novel antimicrobial agents with broad-spectrum activity, such as cefiderocol and plazomicin, are exceptional events, efforts must be made to prevent the dissemination of antimicrobial resistant pathogens. The fact that 50% of the isolates evaluated in this study were recovered from surveillance swabs alerts us to the occurrence of silent reservoirs of NDM-producing *Klebsiella* in hospitalized patients. The effects of increased use of polymyxin and other novel antibiotic combinations (due primarily to the increasing prevalence of other carbapenemase-producing pathogens) on the incidence of metallo-β-lactamase producing organisms in Brazilian settings need to be studied further. Therefore, a real-time, permanent, and nationwide antimicrobial surveillance system is urgently warranted. In summary, our results showed that the occurrence of NDM carbapenemase in *Klebsiella* is driven, in part, by the clonal expansion of the Clonal Group 258, but the non-clonal dispersion was also verified in a great variability of sequence types and even species.

## 4. Materials and Methods

On a continuous and voluntary basis, Instituto Adolfo Lutz receives isolates associated with outbreaks or sporadic cases of infections (especially those with unusual antimicrobial resistance) for phenotypic and genotypic antimicrobial resistance characterization.

Bacterial identification was initially carried out by phenotypic testing, and, for this study, all the isolates were re-identified by MALDI-TOF MS (Bruker Daltonics, Bremen, Germany). Antimicrobial susceptibility testing was performed by disk-diffusion (except for polymyxin B, for which minimal inhibitory concentration was determined by in-house broth microdilution) and breakpoints followed the current EUCAST/BrCAST guidelines (http://brcast.org.br/, accessed on 21 February 2022). The multiple antimicrobial resistance (AMR) index, defined as the ratio of the number of resistances/number of antimicrobials evaluated, was calculated for each isolate. A multiplex PCR targeting the *bla*_KPC_, *bla*_NDM,_ and *bla*_OXA-48_ genes was employed to screen the main carbapenemases found in Enterobacterales [20].

The bacterial genetic diversity was initially assessed by pulsed-field gel electrophoresis (PFGE) following the PulseNet recommendations for plug preparation, enzymatic restriction with XbaI enzyme and running parameters [21]. A dendrogram was built in BioNumerics software v.8.1 (Applied Maths, Sint-Martens-Latem, Belgium) with optimization and tolerance parameters set at 1.5%. Representative isolates from different clusters (preferentially based on a Dice similarity coefficient of 80%) were selected for whole genome sequencing.

Bacterial DNA was extracted by using the Wizard DNA Purification Kit (Promega, Madison, WI, USA) with overnight (24 h) bacterial culture growth on Luria Bertani broth (Difco, Oxford, UK). High quality DNA (assessed by gel electrophoresis and Qubit quantification) was submitted for library preparation with Illumina^®^ DNA Prep Tagmentation and sequenced in a MiSeq (Illumina, San Diego, CA, USA) instrument with MiSeq^®^ Reagent Kit v3 (75 cycles). Library preparation and runs were performed at the Strategic Laboratory, Instituto Adolfo Lutz, Sao Paulo, Brazil.

Quality control was performed with FastQC and Kraken2 software, within the Galaxy Europe Server [22]. The draft genomes were de novo assembled in the software CLC Genomics Workbench (Qiagen, Venlo, The Netherlands) and submitted to PathogenWatch (https://pathogen.watch/upload, accessed on 15 February 2022) [23] for species identification, determination of sequence type (ST), and detection of antimicrobial resistance and virulence genes. The Resfinder, PlasmidFinder (both from the Center for Genomic Epidemiology, https://cge.cbs.dtu.dk/, accessed on 11 April 2022), Comprehensive Antibiotic Resistance Database (CARD, https://card.mcmaster.ca/, accessed on 5 March 2022) [24] and Virulence Factor Database (VFDB, http://www.mgc.ac.cn/VFs/) [25] tools were also employed for genotypic characterization. When necessary, sequences were locally curated into BioNumerics v.8.1 software. All the sequences generated in this study were deposited in the GenBank (BioProject PRJNA879417) and the PubMLST/BigSDB databases (Appendix A).

A phylogeny-based tree was built based on single nucleotide polymorphism (SNPs) using the CSI Phylogeny service from the Center for Genomic Epidemiology (https://cge.cbs.dtu, accessed on 24 February 2022), with default parameters. All the 54 sequences were included in this analysis and the *K. pneumoniae* ATCC BAA 2470 strain (*bla*_NDM-1_-positive, ST37) was used as a reference. The generated tree was uploaded on the Microreact webserver for visualization along with the corresponding metadata [26].

Conjugation experiments were conducted to determine the ability of transference of NDM from the donor bacteria to the *E. coli* J53 recipient strain in representative isolates (with different *bla*_NDM_ alleles and different STs). Separate growths (35 °C, 3 h, agitation of 120 rpm in 3 mL of LB broth) of both isolates were mixed (0.1 mL each) in a tube containing 2 ml of LB broth and incubated overnight at 35 °C without agitation. Transconjugant strains were selected on MacConkey agar with sodium azide (100 µg/mL) and imipenem (1 µg/mL) or ceftazidime (2 µg/mL). Putative transconjugants were reisolated, identified, and submitted to PCR to identify the presence of the *bla*_NDM_ gene. Both the donor and the respective transconjugant isolates were submitted to S1-PFGE in order to identify the size of large plasmids carrying the *bla*_NDM_ gene. NDM activity on transconjugant isolates was evaluated by the antimicrobial susceptibility using the gradient diffusion method (epsilometric strips).

Parental isolates were subjected to library preparation with the rapid BarCoding Sequencing kit and performed in MinIon (Oxford Nanopore, Oxford, UK) to obtain long reads sequencing, aiming to characterize the entire sequence of plasmids carrying the *bla*_NDM_ genes, and assembled with short reads by using Unicycler software [27] following genomic annotation in RAST [28]. Geneious software was employed for data visualization and curation. Plasmid comparison was conducted on BRIG after *blast* analysis [29].

## Figures and Tables

**Figure 1 antibiotics-11-01395-f001:**
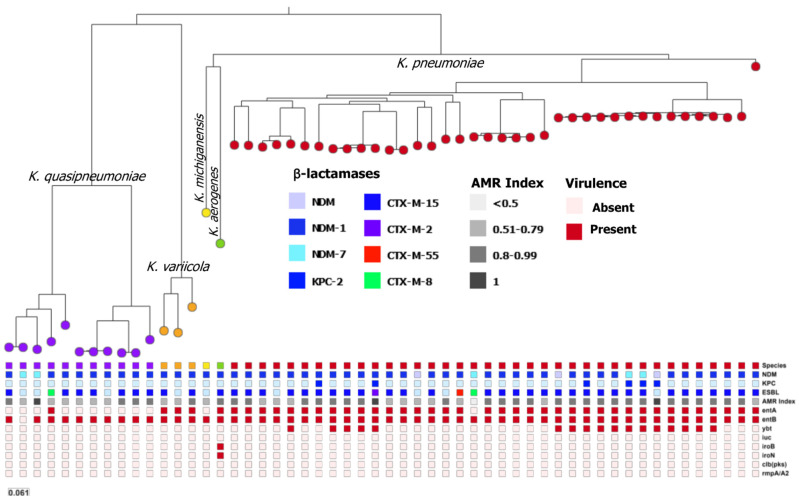
A phylogenetic tree of NDM-producing *Klebsiella* isolates from Brazil. Bacterial identification (species), β-lactamases genes, multiple antimicrobial resistance (AMR) index, and main virulence genes are presented for each isolate. The figure was generated on the Microreact platform based on CSI Phylogeny analysis, using the *K. pneumoniae* ATCC BAA 2470 strain as a reference.

**Figure 2 antibiotics-11-01395-f002:**
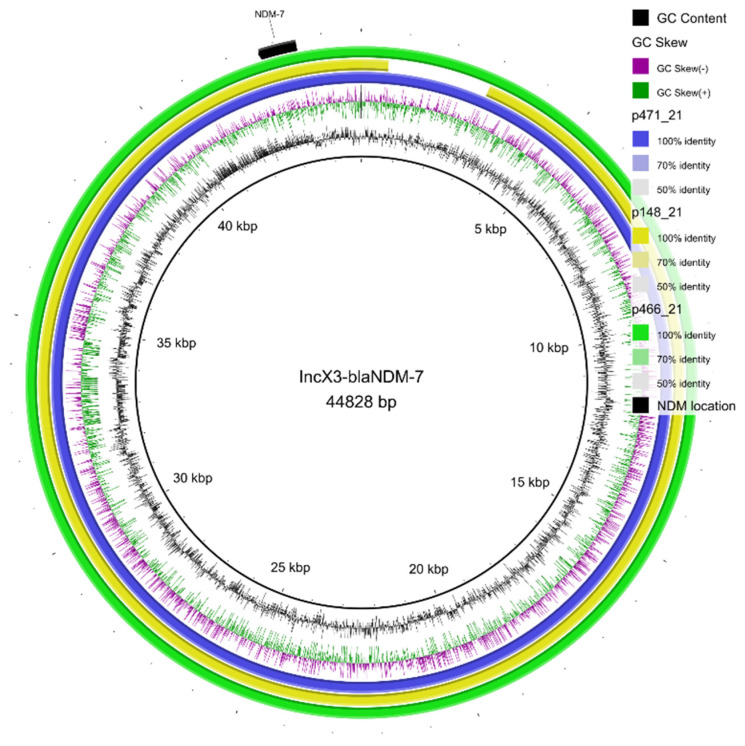
Schematic representation of the IncX3-*bla*_NDM-7_ plasmids identified in *Klebsiella* isolates 471_21 (blue), 148_21 (yellow), and 466_21 (green), highlighting the location of the *bla*_NDM-7_ gene (in black). The figure was generated on BRIG software using p471_21 as a reference.

**Table 1 antibiotics-11-01395-t001:** Antimicrobial susceptibility of NDM-producing *Klebsiella* species from Brazil (n = 135).

Antimicrobial Category	Antimicrobial Agent	Susceptible		Susceptible, Increased Exposure		Resistant	
		n	%	n	%	n	%
Aminoglycosides	Amikacin	76	56.3	0	0.0	59	43.7
	Gentamicin	48	35.6	0	0.0	87	64.4
	Tobramycin	16	11.9	0	0.0	119	88.1
Quinolones	Ciprofloxacin	9	6.7	6	4.4	120	88.9
	Levofloxacin	14	10.4	11	8.1	110	81.5
Folate pathway inhibitors	Trimethoprim- sulfamethoxazole	21	15.6	2	1.5	112	83.0
Penicillin	Ampicillin	0	0.0	0	0.0	135	100.0
Cephalosporins	Cefepime	4	3.0	0	0.0	131	97.0
	Cefotaxime	3	2.2	0	0.0	132	97.8
	Ceftazidime	4	3.0	0	0.0	131	97.0
Carbapenems	Ertapenem	4	3.0	0	0.0	131	97.0
	Imipenem	5	3.7	23	17.0	107	79.3
	Meropenem	5	3.7	10	7.4	120	88.9
Cephamycins	Cefoxitin	2	1.5	0	0.0	133	98.5
β-lactams + β-lactamase inhibitors	Amoxicillin-clavulanic acid	4	3.0	0	0.0	131	97.0
	Ampicillin-sulbactam	3	2.2	0	0.0	132	97.8
	Piperacillin-tazobactam	3	2.2	1	0.7	131	97.0
Monobactam	Aztreonam	11	8.1	2	1.5	122	90.4
Phenicols	Chloramphenicol	54	40.0	0	0.0	81	60.0
Polymyxins	Polymyxin B ^1^	120	88.9	0	0.0	15	11.1

^1^ Polymyxin B MIC50 = 0.5 µg/mL; MIC90 = 4.0 µg/mL.

**Table 2 antibiotics-11-01395-t002:** Antimicrobial susceptibility testing (MIC values, in μg/mL) of NDM-producing transconjugant strains. For comparison, the results for the J53 *E. coli* recipient strain are also presented.

Strain Identification	J53	146_19_trans		148_21_trans		229_19_trans		466_21_trans		471_21_trans	
Parental Species	*E. coli*	*K. pneumoniae*		*K. pneumoniae*		*K. pneumoniae*		*K. pneumoniae*		*K. quasipneumoniae*	
NDM-Type	Recipient Strain	NDM-1		NDM-7		NDM-1		NDM-7		NDM-7	
Antimicrobial Agent	MIC	MIC	X Fold *	MIC	X Fold	MIC	X Fold	MIC	X Fold	MIC	X Fold
Imipenem	0.125	32.0	256	>32.0	>256	12.0	96	>32.0	>256	4.0	32
Meropenem	0.016	4.0	250	6.0	375	3.0	188	>32.0	>2000	1.0	63
Ampicillin	2.0	>256	>128	>256	>128	>256	>128	>256	>128	>256	>128
Cefotaxime	0.015	>256	>17,067	>256	>17,067	256.0	17,067	>256	>17,067	>256	>17,067
Cefepime	0.32	192.0	600	32.0	100	8.0	25	24.0	75	16.0	50
Ceftriaxone	0.015	>32.0	>2133	>32.0	>2133	>32.0	>2133	>32.0	>2133	>32	>2133
Ticarcillin-clavulanic acid	1.5	>256	>171	>256	>171	>256	>171	>256	>171	>256	>171
Cephalotin	3.0	>256	>85	>256	>85	>256	>85	>256	>85	>256	>85

* in comparison with J53 recipient strain.

## Data Availability

All the sequences generated in this study were deposited in the GenBank (BioProject PRJNA879417) and PubMLST/BigSDB databases.

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
