# Peer review of "Genomic Diversity of NDM-Producing Klebsiella Species from Brazil, 2013–2022"

_antibiotics, 2022, doi:10.3390/antibiotics11101395_

Round 1

Reviewer 1 Report

My comments are in the attached file

Author Response

Reviewer 1

Abstract:

Reviewing report

The topic match with the aim of the journal related on and English language and style are fine. This manuscript is about genomic diversity of NDM-producing Klebsiella species  from Brazil, 2013-2022 genetics studies on microorganisms in order to deal with antibiotics resistance and  improve antibiotics use. The article contains interesting data

Author answer: Thank you for the careful evaluation of our study and the suggestions. We hope that the points you raised are now clarified, and the manuscript was improved. All the changes in the text are highlighted in red.

 However, I have some comments that are as follow:

   In the line 15: In this study we evaluated.... It’s unclear, try to explain more precisely  what the aim of the study is

We appreciate your suggestion. The text was changed to: “In this study we characterized by antimicrobial susceptibility testing and molecular typing a large collection of NDM-producing Klebsiella isolates from different hospitals in the state of Sao Paulo, Brazil, over the last decade”

 Introduction:

  The introduction don’t provide sufficient background about epidemiology of NDM- producing Klebsiella in Brasil and its rang about carbapenemase producing klebsiella,  can you develop this?

Thank you for the suggestion. Text was changed to include a brief description of NDM.

The New Delhi Metallo-β-lactamase (NDM) is a potent carbapenemase, initially identified in the Indian subcontinent and now widespread in many countries [5]. In Brazil, the first NDM-producing bacteria was identified in Southern region in 2013 [6], and since then, reported in all the five Brazilian administrative regions from both clinical and surveil-lance specimens [7].

 Results:

  Can you add the clonal distribution of NDM-producing Klebsiella regarding the five Brazilian administrative regions?

Thank you for this suggestion. We included in the text the following sentences, indicating where the isolates came from, and also we decided to include a supplementary table to show this information in more details.

Lines 62-64: These isolates were mainly recovered from hospitals in the states of the Southeast Region (n=109; 80.7%), but also representative from North (n=21; 15.5%), Northeast (n=4; 3.7%), and Midwest (n=1; 0.7%) Regions (Table S1).

Lines 75-76: They were representative of seven different states from four Administrative Regions (Table S1).

Table S1. Distribution of NDM-producing Klebsiella from Brazil, according to state and Administrative Region.

Region/State

Total (n)

%

Sequenced

(n)

%

Sequence types found

Southeast

109

80.7

43

79.6

11 (12), 15 (3), 16, 17, 37, 43, 128, 147, 196 (3), 281, 283 (2), 336, 367, 395, 418, 464, 485, 534, 874, 1040 (2), 1456, 2586, 3128, 4609, 6245 (2)

Rio de Janeiro

1

0.7

1

1.9

464

São Paulo

107

79.3

41

75.9

11 (12), 15 (3), 16, 17, 37, 43, 128, 147, 196 (3), 281, 283 (2), 336, 367, 395, 418, 485, 534, 1040 (2), 1456, 2586, 3128, 4609, 6245 (2)

Minas Gerais

1

0.7

1

1.9

874

Midwest

1

0.7

0

0

-

Goiânia

1

0.7

0

0

-

Northeast

4

3.0

2

3.7

340, 477

Alagoas

1

0.7

1

1.9

477

Pernambuco

3

2.2

1

1.9

340

North

21

15.6

9

16.7

147, 307, 392 (3), 460, 526, 1822, 6244

Pará

8

5.9

2

3.7

147, 1822

Tocantins

13

9.6

7

13.0

307, 392 (3), 460, 526, 6244

Total

135

100

54

100

* In parenthesis are indicated the number of isolates, except when it was 1 (and it was omitted).

 Discussion:

  Add a comparison between the frequency of clonal groups found in your study and  those found in the world

Thank you for this suggestion. We have edited this paragraph in the discussion, and now it reads:

“Globally, NDM is found in diversified bacterial genetic background around the world, including in the epidemic clones K. pneumoniae ST11 and ST147 [5,9]. In this study, we identified globally recurrent clones (ST11, ST147, ST340, and ST15) in our isolates, reflecting the dynamics of antimicrobial resistance observed abroad [9]. The CG258 (which comprises ST11, ST340) is one of the most disseminated clone worldwide, usually associated with multidrug antimicrobial resistance [10].”

 Material and methods:

  Add references of genomics data bases in the references part

Thank you for the observation. Some tools are cited only by the website address, such as the CSI Phylogeny tool. For the remaining softwares, the references were included:

  1. Argimón, S.; David, S.; Underwood, A.; Abrudan, M.; Wheeler, N.E.; Kekre, M.; Abudahab, K.; Yeats, C.A.; Goater, R.; Taylor, B.; et al. Rapid Genomic Characterization and Global Surveillance of Klebsiella Using Pathogenwatch. Clin. Infect. Dis. 2021, 73, S325–S335, doi:10.1093/CID/CIAB784.
  2. Alcock, B.P.; Raphenya, A.R.; Lau, T.T.Y.; Tsang, K.K.; Bouchard, M.; Edalatmand, A.; Huynh, W.; Nguyen, A.L. V.; Cheng, A.A.; Liu, S.; et al. CARD 2020: Antibiotic Resistome Surveillance with the Comprehensive Antibiotic Resistance Database. Nucleic Acids Res. 2020, 48, D517–D525, doi:10.1093/nar/gkz935.
  3. Chen, L.; Yang, J.; Yu, J.; Yao, Z.; Sun, L.; Shen, Y.; Jin, Q. VFDB: A Reference Database for Bacterial Virulence Factors. Nucleic Acids Res. 2005, 33, doi:10.1093/NAR/GKI008.
  4. Argimón, S.; Abudahab, K.; Goater, R.J.E.; Fedosejev, A.; Bhai, J.; Glasner, C.; Feil, E.J.; Holden, M.T.G.; Yeats, C.A.; Grundmann, H.; et al. Microreact: Visualizing and Sharing Data for Genomic Epidemiology and Phylogeography. Microb. genomics 2016, 2, e000093, doi:10.1099/mgen.0.000093.

Reviewer 2 Report

Comment and suggestion to authors:

Manuscript ID: antibiotics-1943241

Type: Article

Titled:  Genomic diversity of NDM-producing Klebsiella species from Brazil, 2013-2022

1) As Antibiotics is Q1 journal with good Impact Factor, and the content of this manuscript is quite short and present the result about collection of NDM-producing Klebsiella isolates from different hospitals in the state of Sao Paulo. So, I would like to suggest to change the type of this manuscript to be “communication”

2) The Figure 2 should be revised, because the text in the figure are very thin and small.

3) Line 88-99, the results about phylogenetic tree based on single nucleotide polymorphisms (SNP) should be explained in more detail and compared with the others studies.

4) The greater number of other related published works should be added to discuss with the results from this current study.

5) It would also very nice, if the authors can provide the additional future perspective about the application of their results into the end of the discussion section.

6)There are some spelling mistakes and grammatical error found in this manuscript.

Author Response

Reviewer 2 (R2)

1) As Antibiotics is Q1 journal with good Impact Factor, and the content of this manuscript is quite short and present the result about collection of NDM-producing Klebsiella isolates from different hospitals in the state of Sao Paulo. So, I would like to suggest to change the type of this manuscript to be “communication”

Thank you so much for your time in reading our manuscript so carefully. We do appreciate your suggestion, but we summarized our results to fit the article format. I hope that the reviewer can understand that the number of pages and the references is not suitable to publish it as a Communication. We present a revised version, and we believe that now it is ready to be published for Antibiotics journal readers.

2) The Figure 2 should be revised, because the text in the figure are very thin and small.

We appreciate your suggestion. A file in higher resolution (300dpi) was submitted in this version. Since it was generated in BRIG, some configurations cannot be changed. Hope that now the figure is more readable.

3) Line 88-99, the results about phylogenetic tree based on single nucleotide polymorphisms (SNP) should be explained in more detail and compared with the others studies.

Pairwise comparison of each genome identified that the SNPs variation ranged from 1 to 50489. We observed a main cluster comprising the K. pneumoniae isolates with SNPs ranging from 1 to 9639. The K. quasipneumoniae cluster was subdivided in two subclusters with SNPs ranging from 11 to 9186 and from 78 to 9070, respectively; overall distance among K. quasipneumoniae was 29757 SNPs. The highest diversity was observed for K. variicola, with SNPs number ranging from 9185 to 9899.

Comparison with other studies is quite difficult to be made, since this approach is more useful for comparing a well defined dataset; even the reference choice impact on SNPs analysis and this can lead to misinterpretation of results.

4) The greater number of other related published works should be added to discuss with the results from this current study.

Thank you for this observation. We have included some references that summarize the current status of NDM-producing bacteria both globally and a recent review from Brazil. The text was changed to address the status of the clones found in our study in the light of the current knowledge.

“In addition, this study identified other 14 ST that have never been associated with NDM carbapenemase in our country, as summarized by a recent review [7], providing an update to the molecular epidemiological scenario of NDM-producing K. pneumoniae in Brazil (…)”

“Globally, NDM is found in diversified bacterial genetic background around the world, including in the epidemic clones K. pneumoniae ST11 and ST147 [5,9]. In this study, we identified globally recurrent clones (ST11, ST147, ST340, and ST15) in our isolates, re-flecting the dynamics of antimicrobial resistance observed abroad [9].”

5) It would also very nice, if the authors can provide the additional future perspective about the application of their results into the end of the discussion section.

We appreciate your suggestion, and we included a sentence in the end of the discussion, summarizing the study results.

“In summary, our results showed that occurrence of NDM carbapenemase in Klebsiella is driven, in parts, by the clonal expansion of the Clonal Group 258, but the non-clonal dispersion was also verified in a great variability of sequence types and even species.”

6)There are some spelling mistakes and grammatical error found in this manuscript.

Thank you for the time spent in reading our manuscript. We have revised the entire text in order to correct those mistakes.

Round 2

Reviewer 2 Report

Manuscript ID: antibiotics-1943241

Type: Article

Titled:  Genomic diversity of NDM-producing Klebsiella species from Brazil, 2013-2022

        Thanks to share the revised version and address to my comments. Comparing with the current publications in Antibiotics, I still recommend to publish this manuscript in the form of “communication” as I already suggested in the first round of revision.

       “As Antibiotics is Q1 journal with good Impact Factor, and the content of this manuscript is quite short and present the result about collection of NDM-producing Klebsiella isolates from different hospitals in the state of Sao Paulo. So, I would like to suggest to change the type of this manuscript to be “communication” ”

               In addition, it is nice to see that the authors add 3 references to discuss. But it would be very nice to add some more recent publications that related to this study to discuss with the results from this current study. It must be very nice for Antibiotics’ s readers.

Author Response

Answer to reviewer 2:

  Thanks to share the revised version and address to my comments. Comparing with the current publications in Antibiotics, I still recommend to publish this manuscript in the form of “communication” as I already suggested in the first round of revision.

       “As Antibiotics is Q1 journal with good Impact Factor, and the content of this manuscript is quite short and present the result about collection of NDM-producing Klebsiella isolates from different hospitals in the state of Sao Paulo. So, I would like to suggest to change the type of this manuscript to be “communication” ”

We are profoundly grateful for this second round of revision of our study. We apologize not attendind this suggestion in the first review. Now we present our manuscript as a Communication. We hope that it could be informative to Antibiotics’ readers.

               In addition, it is nice to see that the authors add 3 references to discuss. But it would be very nice to add some more recent publications that related to this study to discuss with the results from this current study. It must be very nice for Antibiotics’ s readers.

We agree with the reviewer, and, as requested, we detailed a little bit more our discussion. We included information to put our results in a global context, including information on the main clones found in Brazil, as well as the reported ST identified worldwide. In addition, we identified that some ST found in our study have been associated with outbreaks before, and we decided to include also this information and reference in this version. Moreover, we felt that we could discuss the plasmids backbone found in our study in a wider context, and this sentence was also included. All the changes are pointed below, and are highlighted in red in the manuscript.

In a recent narrative review of NDM-producing Enterobacterales in Brazil, we identified that blaNDM-1 is mainly identified in Klebsiella species, belonging to ST350 and ST15 [1].

Among the 21 ST identified in our study, at least seven of them have already been associated with outbreaks worldwide: ST11 (Bulgaria, Greece, and Turkey), ST15 (Nepal), ST17 and ST37 (China), ST147 (China and Tunisia), ST392 (Mexico) and ST340, which caused a large outbreak in two Brazilian hospitals [2,3]; these findings indicate that occasional clones have the potential to cause outbreaks under favorable conditions.

Contrariwise, blaNDM-1 was found to be located into IncF plasmids in our isolates. A recent study found the occurrence of blaNDM-1 in a megaplasmid carried by a K. pneumoniae ST1588 recovered from rectal swab in a patient from Chile [4]. Since 2011 a variety of plasmids have been found to be associated with NDM gene, such as the incompatibility (Inc) groups IncF, IncA/C, IncL/M, or even in the bacterial chromosome [5]. In Brazil, blaNDM gene was also observed in plasmids with different sizes [6], indicating that this resistance determinant occurs in different mobile genetic elements backbones [2].

References

  1. Camargo, C.H. Current Status of NDM-Producing Enterobacterales in Brazil: A Narrative Review. Brazilian J. Microbiol. 2022, doi:10.1007/s42770-022-00779-1.
  2. Wu, W.; Feng, Y.; Tang, G.; Qiao, F.; McNally, A.; Zong, Z. NDM Metallo-β-Lactamases and Their Bacterial Producers in Health Care Settings. Clin. Microbiol. Rev. 2019, 32.
  3. Monteiro, J.; Inoue, F.M.; Lobo, A.P.T.; Ibanes, A.S.; Tufik, S.; Kiffer, C.R.V. A Major Monoclonal Hospital Outbreak of NDM-1-Producing Klebsiella Pneumoniae ST340 and the First Report of ST2570 in Brazil. Infect. Control Hosp. Epidemiol. 2019, 40, 492–494, doi:10.1017/ice.2018.333.
  4. Quezada-Aguiluz, M.; Opazo-Capurro, A.; Lincopan, N.; Esposito, F.; Fuga, B.; Mella-Montecino, S.; Riedel, G.; Lima, C.A.; Bello-Toledo, H.; Cifuentes, M.; et al. Novel Megaplasmid Driving NDM-1-Mediated Carbapenem Resistance in Klebsiella Pneumoniae ST1588 in South America. Antibiotics 2022, 11, 1207, doi:10.3390/antibiotics11091207.
  5. Poirel, L.; Dortet, L.; Bernabeu, S.; Nordmann, P. Genetic Features of BlaNDM-1-Positive Enterobacteriaceae. Antimicrob. Agents Chemother. 2011, 55, 5403–5407, doi:10.1128/AAC.00585-11.
  6. Rozales, F.P.; Magagnin, C.M.; Campos, J.C.; Pagano, M.; Nunes, L.S.; Pancotto, L.R.; Sampaio, J.L.M.; Zavascki, A.P.; Barth, A.L. Characterization of Transformants Obtained from NDM-1-Producing Enterobacteriaceae in Brazil. Infect. Control Hosp. Epidemiol. 2017, 38, 634–636.